# Do We Need Neural Collapse? Learning Diverse Features for Fine-grained and Long-tail Classification

## Abstract

Feature extractors *learned* from supervised training of deep neural networks have demonstrated superior performance over *handcrafted* ones. Recently, it is shown that such learned features have a *neural collapse* property, where within-class features collapse to the class mean and different class means are maximally separated. This paper examines the neural collapse property in the context of fine-grained classification tasks, where a feature extractor pretrained from a classification task with *coarse* labels is used for generating features for a downstream classification task with *fine-grained* labels. We argue that the within-class feature collapse is an undesirable property for fine-grained classification. Hence, we introduce a geometric arrangement of features called the *maximal-separating-cone*, where within-class features lie in a cone of nontrivial radius instead of collapsing to the class mean, and cones of different classes are maximally separated. We present a technique based on classifier weight and training loss design to produce such an arrangement. Experimentally we demonstrate an improved fine-grained classification performance with a feature extractor pretrained by our method. Moreover, our technique also provides benefits for classification on data with long-tail distribution over classes. Our work may motivate future efforts on the design of better geometric arrangement of deep features.

## 1 Introduction

The extraction of *features* or *representations* from image, language, and speech data in their raw forms is a problem of fundamental interest in machine learning. A standard approach in classical machine learning is to handcraft a feature extractor that maps from the input to the feature space (Lowe, 2004), which often requires meticulous and onerous work of human and domain experts. With the success of deep learning, methods based on *learned* feature extractors have become very popular. Such an approach not only requires less human expertise, but also exhibits better empirical performance compared to handcrafted feature extractors (Krizhevsky et al., 2017). Moreover, deep features extractors obtained from a pre-training task can be used for downstream tasks by simply training a task-specific classifier, which offers good empirical performance (Yosinski et al., 2014).

The great success of learned features naturally leads to the following question: Are learned features *optimal* for various application scenarios? In this paper, we show that learned features in a standard deep classification model are not optimal and can be improved by handcrafting a better geometric arrangement of the features. We are motivated by a recent line of work that shows that the geometric arrangement of features in a standard deep classification model has a simple and elegant form:

- Within each class, the features are maximally concentrated and collapse to the class mean.
- Across different classes, the features are maximally separated. Namely, the distance between each pair of class means is the same and that distance is at the maximum.

This phenomenon is called *Neural Collapse ($\mathcal{NC}$)* (Papyan et al., 2020). The occurrence and prevalence of $\mathcal{NC}$ is verified empirically through experiments with a variety of datasets and network architectures (Han et al., 2021). Motivated by such an observation, recent work provided theoretical analysis of deep features under the so-called *unconstrained feature models* (Mixon et al., 2020;

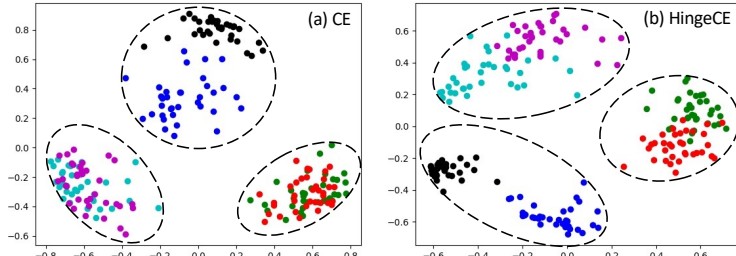

Figure 1: Fine-grained classification with $\mathcal{NC}$ vs. diverse features. We train a ResNet32 on CIFAR-20, which contains 20 coarse classes, and use the features for classification on CIFAR-100, which contains 100 fine-grained class labels of the 20 coarse classes. The plots provide a visualization of randomly sampled features from three selected coarse classes (enclosed in dashed circles), each with two selected fine-grained classes (shown with different colors). Our visualization uses the technique of Müller et al. (2019). (a) With cross-entropy (CE) loss, within-class features collapse and fine-grained classes are less distinguishable. (b) With our method (i.e., Hinge CE), within-class features are diverse and fine-grained classes are more distinguishable.

Fang et al., 2021), where the features are considered to be free optimization variables for the reason that deep neural networks have sufficient expressive power to produce any features for any training dataset. In particular, Lu & Steinerberger (2020); Weinan & Wojtowytsch (2020); Mixon et al. (2020); Graf et al. (2021); Fang et al. (2021); Ji et al. (2021); Tirer & Bruna (2022) show that $\mathcal{NC}$ solutions are global optimal solutions for the nonconvex objective functions. Moreover, Zhu et al. (2021); Zhou et al. (2022) show that such nonconvex problems have a benign global landscape hence $\mathcal{NC}$ can be easily obtained by standard training algorithms. The study of $\mathcal{NC}$ arguably brings us a better understanding of the properties of the deep features. Nonetheless, existing study does not provide an answer to the following question:

*Is Neural Collapse the desired property for deep features?*

**Contribution.** We provide a study of whether $\mathcal{NC}$ is the desired property for deep features by considering a specific learning task. In particular, we consider the fine-grained classification task, where a deep learning model is first pretrained on a classification dataset with *coarse* class labels, and use the features extracted from such a model for a downstream classification task with *fine-grained* classes. For example, the pretraining task contains a coarse class, say *flower*, that contains all variants of flowers, while the fine-grained task may give each variant of flower a different class label. If $\mathcal{NC}$ occurs hence all features for the coarse class collapse to the class mean, then such features do not distinguish between different fine-grained classes hence cannot be used to successfully perform fine-grained classification. This simple thought experiment demonstrates that $\mathcal{NC}$ is not a desired property for fine-grained classification.

Motivated by the observation above, we argue that deep features extracted during pretraining should not have the $\mathcal{NC}$ property for the task of fine-grained classification. Instead, the features for each pretraining class should be *diverse* to a certain extent, so that variation within the class is preserved and can benefit fine-grained classification. To realize this idea, we introduce a geometric arrangement of deep features called the *maximal-separating-cone* (MSC), where features of each class lie inside a *cone* with a nontrivial radius (i.e. instead of collapsing to class mean as in $\mathcal{NC}$), and the axes of the cones are maximally separated. See Figure 2 for an illustration. To obtain such an arrangement, we present a technique based on the simple idea that, if a feature lies inside its cone, then its loss is set to be a constant by a hinge function. Figure 1 provides a comparison of $\mathcal{NC}$ features and our MSC features for fine-grained classification with CIFAR-20 and CIFAR-100. The contribution of this work is summarized as follows.

- We design a MSC arrangement of deep features where within-class features are diverse within a cone instead of collapsing to the class mean (as in $\mathcal{NC}$).

- We present a generic technique based on the design of classifier weight and training loss function to obtain MSC features.

- We conduct experiments on fine-grained classification to demonstrate the effectiveness of our method. We also provide ablation study to justify the design of each component in our method.

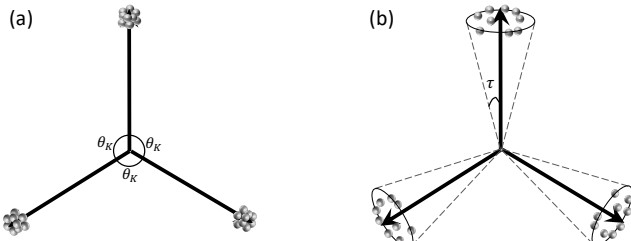

Figure 2: Conceptual illustration of features obtained from deep learning based classification models. Each ball represents a feature vector for an input in the training set. (Left) Neural collapse features obtained by standard training ( Section 2.2), where within each class, all features collapse to the class mean. (Right) Maximal-separating-cone features that our method obtains (Section 3.1), where within each class, the features lie in a cone and the axes of the cones are maximally separated.

- Aside from fine-grained classification, we demonstrate that our method also improves the performance for classification with training data that has imbalanced number of samples for different classes (i.e., long-tail data).

## 2 BACKGROUND

In this section, we start by reviewing the basics of deep learning based classification in Section 2.1, where a feature extractor parameterized by a deep neural network and a linear classifier on the features are jointly learned from a training dataset. Then, Section 2.2 explains $\mathcal{NC}$ which characterizes the features and classifier obtained from training a deep learning based classification model.

### 2.1 DEEP LEARNING BASED CLASSIFICATION

Consider the classification task where the goal is to predict the label $y \in [K] \doteq \{1, \ldots, K\}$ for an input $\boldsymbol{x} \in \mathbb{R}^n$, where $K$ is the number of classes. With deep learning based classification, this is typically achieved by training a model with the following optimization objective:

$$\min_{\boldsymbol{\theta}, \boldsymbol{W}, \boldsymbol{b}} \frac{1}{N} \sum_{i=1}^{N} \mathcal{L}\big([v_{i1}, \ldots, v_{iK}], y_i\big), \text{ s.t. } \boldsymbol{u}_i = f(\boldsymbol{x}_i, \boldsymbol{\theta}), \ v_{ik} = \langle \boldsymbol{u}_i, \boldsymbol{w}_k \rangle + b_k, \quad (1)$$

where $\{\boldsymbol{x}_i, y_i\}_{i=1}^{N}$ is a training dataset of input-label pairs. In above, $f(\cdot, \boldsymbol{\theta}) : \mathbb{R}^n \to \mathbb{R}^m$ is the feature extractor implemented as a deep neural network with trainable parameters $\boldsymbol{\theta}$. The output $\boldsymbol{u}_i$ of the feature extractor is referred to as the feature of $\boldsymbol{x}_i$. The other two optimization variables in (1), namely $\boldsymbol{W} = [\boldsymbol{w}_1, \ldots, \boldsymbol{w}_K] \in \mathbb{R}^{m \times K}$ and $\boldsymbol{b} = [b_1, \ldots, b_K] \in \mathbb{R}^K$, are parameters of a linear classifier. The vector $[v_{i1}, \ldots, v_{iK}]$ is often referred to the *logit* vector for data $\boldsymbol{x}_i$. The loss function $\mathcal{L}(\cdot, \cdot)$ is defined on the logits and the label. One of the most popular choice of $\mathcal{L}(\cdot, \cdot)$ is the cross-entropy (CE) loss, but other choices such as the mean squared error loss may also induce $\mathcal{NC}$.

### 2.2 NEURAL COLLAPSE ($\mathcal{NC}$)

$\mathcal{NC}$ is a phenomenon for the features obtained at terminal phase of training a deep learning based classification model (Papyan et al., 2020). Let $(\boldsymbol{\theta}^*, \boldsymbol{W}^*, \boldsymbol{b}^*)$ be the solution to (1). Then, $\mathcal{NC}$ states that the learned features $\{\boldsymbol{u}_i^* \doteq f(\boldsymbol{x}_i, \boldsymbol{\theta}^*)\}_{i=1}^{N}$ have the following properties (see also Figure 2(a)).

(i) $\mathcal{NC}$**1: Within-class Variability Collapse.** For each class $k \in [K]$, the features $\{\boldsymbol{u}_i^* \mid i : y_i = k\}$ collapse to the class mean $\boldsymbol{\mu}_k^* \doteq \frac{\sum_{i:y_i=k} \boldsymbol{u}_i^*}{\sum_{i:y_i=k} 1}$.

(ii) $\mathcal{NC}$**2: Between-class Maximal Separation.** Let $\boldsymbol{\mu}_G^* = \frac{1}{K} \sum_{k=1}^{K} \boldsymbol{\mu}_k^*$ be the global mean. Then, the class means after subtracting the global mean, i.e., $\{\boldsymbol{\mu}_k^* - \boldsymbol{\mu}_G^* \mid k \in [K]\}$ are maximally distant and located on a sphere centered at origin. More precisely, they form a simplex equiangular tight frame (Simplex ETF) up to a rotation and scaling:

$$[\boldsymbol{\mu}_1^* - \boldsymbol{\mu}_G^*, \ldots, \boldsymbol{\mu}_K^* - \boldsymbol{\mu}_G^*] = s\boldsymbol{U}\boldsymbol{W}_{\text{ETF}}, \quad (2)$$

where $s > 0$ is an arbitrary scaling and $\boldsymbol{U} \in \mathbb{R}^{m \times K}$ is an arbitrary rotation (i.e., orthonormal matrix)[1] of the Simplex ETF. In above, $\boldsymbol{W}_{\text{ETF}}$ is a matrix that defines a Simplex ETF:

$$\boldsymbol{W}_{\text{ETF}} \doteq \sqrt{\frac{K}{K-1}} \Big(\mathbf{I}_K - \frac{1}{K} \mathbf{1}_K \mathbf{1}_K^T\Big) \tag{3}$$

where $\mathbf{I}_K \in \mathbb{R}^{K \times K}$ is an identity matrix and $\mathbf{1}_K \in \mathcal{R}^K$ is a vector of all ones. It can be verified that the angle between any pair of class means after subtracting the global mean is the same:

$$\arccos\left(\frac{\langle \boldsymbol{\mu}_k^* - \boldsymbol{\mu}_G^*, \ \boldsymbol{\mu}_{k'}^* - \boldsymbol{\mu}_G^* \rangle}{\|\boldsymbol{\mu}_k^* - \boldsymbol{\mu}_G^*\|_2 \cdot \|\boldsymbol{\mu}_{k'}^* - \boldsymbol{\mu}_G^*\|_2}\right) = \boldsymbol{\theta}_K \doteq \arccos\left(-\frac{1}{K-1}\right), \forall k, k' \in [K], k \neq k'. \tag{4}$$

Aside from the learned features, $\mathcal{NC}$ also states the following two properties hold for the classifier.

(iii) $\mathcal{NC}$**3: Self-duality between Feature and Classifier.** The classifier weight $\boldsymbol{W}^* = [\boldsymbol{w}_1^*, \ldots, \boldsymbol{w}_K^*]$ satisfies $\boldsymbol{w}_k^* = C \cdot (\boldsymbol{\mu}_k^* - \boldsymbol{\mu}_G^*)$, where $C$ is a scalar independent of $k$.

(iv) $\mathcal{NC}$**4: Nearest Class Mean Classifier.** The class label for a test data $\boldsymbol{x}$ is determined by a nearest class mean classifier: $\arg\min_{k \in [K]} \|f(\boldsymbol{x}, \boldsymbol{\theta}^*) - \boldsymbol{\mu}_k^*\|_2$.

Theoretical studies of the $\mathcal{NC}$ phenomenon are often conducted with the unconstrained feature models (UFMs). The idea of UFMs is that, since performing an analysis of a multi-layer nonlinear neural network $f(.)$ is difficult, we treat $f(.)$ as a model that can produce any set of features given any set of inputs. Hence, the features $\{\boldsymbol{u}_i\}_{i=1}^K$ can be treated as free optimization variables, which allows us to peel off the network $f(.)$ and consider the following optimization problem[2]

$$\min_{\{\boldsymbol{u}_i\}_{i=1}^N, \boldsymbol{W}, \boldsymbol{b}} \frac{1}{N} \sum_{i=1}^N \mathcal{L}\big([v_{i1}, \ldots, v_{iK}], y_i\big), \ \ \text{s.t.} \ \ v_{ik} = \langle \boldsymbol{u}_i, \boldsymbol{w}_k \rangle + b_k. \tag{5}$$

Under certain assumptions, it can be shown that all global solutions to (5) satisfy $\mathcal{NC}$ and that such an optimization problem has a benign global landscape (Zhu et al., 2021).

## 3 METHOD

In this section, we start by presenting an arrangement of features described by their geometric properties in Section 3.1, where within-class features are diverse within a cone. Then, Section 3.2 introduces our method for obtaining such an arrangement by the design of classifier and loss function.

### 3.1 A MAXIMAL-SEPARATING-CONE (MSC) ARRANGEMENT

As explained in Section 2.2, a standard deep learning based classification produces features with an arrangement described by $\mathcal{NC}$, where within-class variability converges to zero (i.e., $\mathcal{NC}$1) and between-class separation is maximized (i.e., $\mathcal{NC}$2) (see Figure 2(a)). Here we introduce an arrangement of features where within-class features are allowed to be diverse. As briefly explained in the introduction, within-class diversity is a desirable property for fine-grained classification.

**Definition 1** (Maximal-Separating-Cone Arrangement). *The set of features $\{\boldsymbol{u}_i\}_{i=1}^N \subseteq \mathbb{R}^m$ is said to have a maximal-separating-cone (MSC) arrangement with respect to (w.r.t.) the set of labels $\{y_i\}_{i=1}^N \subseteq [K]$ with angle $\tau > 0$ if there exist a collection of vectors $\{\boldsymbol{\beta}_k\}_{k=1}^K \subseteq \mathbb{R}^m$ with $\{\boldsymbol{\beta}_k\}_{k=1}^K$ forming a Simplex ETF, such that for all $i, k$ that satisfies $y_i = k$, it has*

$$\frac{\langle \boldsymbol{u}_i, \ \boldsymbol{\beta}_k \rangle}{\|\boldsymbol{u}_i\|_2 \cdot \|\boldsymbol{\beta}_k\|_2} \geq \cos \tau. \tag{6}$$

The MSC arrangement has a simple geometric description (see Figure 2(b)). In words, it requires that features associated with each class lies in a cone. Moreover, the cones have the properties that 1) their axes form a Simplex ETF, hence different classes are sufficiently separated, and that 2) their angular radius is $\tau > 0$, hence within-class features do not collapse to a single point as in $\mathcal{NC}$.

---

[1] We assume that the feature space dimension $m$ is greater or equal to the number of classes $K$.

[2] A regularization term is often added to the optimization variables. We omit it to simplify the discussion.

## 3.2 Inducing MSC Features via Classifier and Loss Design

We obtain MSC features by training a deep neural network with the following optimization problem:

$$\min_{\boldsymbol{\theta}} \frac{1}{N} \sum_{i=1}^{N} \mathcal{L}\Big([v_{i1}, \ldots, v_{iK}], y_i\Big), \ \ \text{s.t.} \ \ \boldsymbol{u}_i = f(\boldsymbol{x}_i, \boldsymbol{\theta}), \ \ v_{ik} = h(\boldsymbol{u}_i, y_i, \boldsymbol{w}_k, b_k),$$

$$\boldsymbol{W} \doteq [\boldsymbol{w}_1, \ldots, \boldsymbol{w}_K] = \boldsymbol{W}_{\text{ETF}}, \ \text{and} \ \boldsymbol{b} \doteq [b_1, \ldots, b_K] = \boldsymbol{0},$$

(7)

where $h(\cdot)$ is defined in (8) and $\boldsymbol{W}_{\text{ETF}}$ is a Simplex ETF matrix defined in (3). The problem in (7) differs from (1) in two aspects, which we explain below.

**Fixed Classifier.** Instead of learning the classifier weights $\boldsymbol{W}$ and $\boldsymbol{b}$ from training data, we fix them as constant model parameters in (7). In particular, $\boldsymbol{W}$ is fixed as a Simplex ETF matrix $\boldsymbol{W}_{\text{ETF}}$.[3] The idea is that we would like the columns of $\boldsymbol{W}$ to be the axes of the cones in MSC arrangement, which are required to form a Simplex ETF according to Definition 1. For the parameter $\boldsymbol{b}$, we simply fix it to be a zero vector, which makes the design of method for obtaining MSC features easier.

**Hinge Loss.** The second difference in (7) is that the logits are set as $v_{ik} = h(\boldsymbol{u}_i, y_i, \boldsymbol{w}_k, b_k)$ with $h(.)$ being a hinge function defined below:

$$h(\boldsymbol{u}_i, y_i, \boldsymbol{w}_k, b_k) = \begin{cases} h^+(\boldsymbol{u}_i, \boldsymbol{w}_k, b_k) \doteq \min\{\langle \boldsymbol{w}_k, \boldsymbol{u}_i \rangle, \text{stop-grad}(\|\boldsymbol{u}_i\|_2)\|\boldsymbol{w}_k\|_2 \cos \tau^+\} + b_k, & \text{if } k = y_i, \\ h^-(\boldsymbol{u}_i, \boldsymbol{w}_k, b_k) \doteq \max\{\langle \boldsymbol{w}_k, \boldsymbol{u}_i \rangle, \text{stop-grad}(\|\boldsymbol{u}_i\|_2)\|\boldsymbol{w}_k\|_2 \cos \tau^-\} + b_k, & \text{if } k \neq y_i, \end{cases}$$

(8)

where $\tau^+$ and $\tau^-$ are two hyper-parameters. If $\tau^+ = 0$ and $\tau^- = \pi$, we obtain $h(\boldsymbol{u}_i, y_i, \boldsymbol{w}_k, b_k) = \langle \boldsymbol{w}_k, \boldsymbol{u}_i \rangle + b_i$ which is the logit function used in (7). The idea of designing (8) is explained below.

- For the logit $v_{ik}$ corresponding to the positive class, i.e., when $k = y_i$, we define $v_{ik} = \langle \boldsymbol{w}_k, \boldsymbol{u}_i \rangle$, which is the typically used logit in (7), when the angle between $\boldsymbol{w}_k$ and $\boldsymbol{u}_i$ is greater than $\tau^{+}$.[4] Otherwise, we define the logit to be $v_{ik} = \text{stop-grad}(\|\boldsymbol{u}_i\|_2) \cos \tau^+$, with stop-grad() denotes an operation to stop the backpropagation of gradient; thereby making the logit independent of network parameters $\boldsymbol{\theta}$.
- For logits $v_{ik}$ corresponding to the negative classes, i.e., when $k \neq y_i$, we define $v_{ik} = \langle \boldsymbol{w}_k, \boldsymbol{u}_i \rangle$, which is the typically used logit in (7), if the angle between $\boldsymbol{w}_k$ and $\boldsymbol{u}_i$ is smaller than $\tau^-$. Otherwise, we define $v_{ik} = \text{stop-grad}(\|\boldsymbol{u}_i\|_2) \cos \tau^-$, so that the logits becomes independent of network parameters $\boldsymbol{\theta}$.

By setting $\tau^+ = \tau$, the logit $v_{ik}$ for the positive class is independent of network parameters $\boldsymbol{\theta}$ when $\boldsymbol{u}_i$ lies in its corresponding cone in an MSC. Similarly, using the fact that the pairwise angular distance of vectors in a Simplex ETF with $K$ vectors is the same and is given by $\boldsymbol{\theta}_K$ (see (4)), by setting $\tau^- = \theta_K - \tau$, the logits $v_{ik}$ for the negative classes are independent of $\boldsymbol{\theta}$ when $\boldsymbol{u}_i$ lies in its corresponding cone in an MSC. Combining the cases above, we readily obtain the following result.

**Theorem 1.** *Consider* (7) *with* $\tau^+ = \boldsymbol{\theta}_K - \tau^- = \tau$. *If the set of features* $\{\boldsymbol{u}_i\}_{i=1}^N$ *satisfy the MSC arrangement w.r.t.* $\{y_i\}_{i=1}^N$ *with angle* $\tau$, *then the gradient of the objective w.r.t.* $\boldsymbol{\theta}$ *is zero.*

From Theorem 1, if the features already have an MSC arrangement, then the network $f()$ will no longer be updated and hence the features will not collapse to their corresponding class centers.

## 4 Experiments

In this section, we demonstrate the empirical performance of our approach. In particular, we focus on two applications: (i) *Fine-grained classification*, where the goal is to illustrate the benefit of diverse features learned during pre-training for fine-grained downstream classification, and (ii) *Long-tail classification*, where we apply our approach to improve the performance on tail classes.

For our approach, we train feature extractors $f()$ by solving the optimization problem in (7) with $\mathcal{L}$ being the CE loss and $f()$ being different variants of ResNet (He et al., 2016) depending on the

---

[3]This design choice was previously made in Zhu et al. (2021), with the idea that since $\mathcal{NC}$ occurs at convergence, simply fixing $\boldsymbol{W}$ to be a a Simplex ETF should not hurt performance while may accelerate training. More broadly, Hoffer et al. (2018) also noted that the classifier need not be trained.

[4]Here we use the fact that $\|\boldsymbol{w}_k\|_2 = 1$ by the constraint in (7).

Table 1: Performance comparison of transfer learning using linear evaluation on nine downstream image classification datasets. We report the Top-1 accuracy (%) where the **first** and the second high results are highlighted. Our approach, HingeCE, improves the transfer learning accuracy consistently and reaches the highest mean accuracy on several datasets. Symbols $\dagger$ & * here indicate the results reported in Chen et al. (2020) and Kornblith et al. (2021) respectively.

| Approach | Food | Cifar10 | Cifar100 | Birdsnap | SUN | Car | Aircraft | Pet | Flower | Mean |
|---|---|---|---|---|---|---|---|---|---|---|
| SimCLR$^\dagger$ | 68.4 | 90.6 | 71.6 | 37.4 | 58.8 | 50.3 | 50.3 | 83.6 | 91.2 | 66.9 |
| CE* | 74.6 | **92.4** | **76.9** | 55.4 | 62.0 | 60.3 | 60.7 | 92.0 | 94.0 | 74.3 |
| Smooth* | 72.7 | 91.6 | 75.2 | 53.6 | 61.6 | 54.8 | 58.9 | **92.9** | 91.9 | 72.6 |
| CE+ETF | 73.4 | 91.0 | 73.0 | 55.1 | 65.3 | 60.1 | 59.9 | 92.0 | 95.8 | 74.0 |
| Smooth+ETF | 70.5 | 90.6 | 71.4 | 54.6 | 64.7 | 58.8 | 59.0 | 90.2 | 94.3 | 72.7 |
| HingeCE (Ours) | **74.9** | 92.3 | 76.6 | **55.8** | **68.2** | **62.2** | **61.2** | 92.2 | **97.8** | **75.7** |

dataset and application. We refer to our method as *HingeCE*. Since the last two layers of (all variants of) ResNets is a ReLU activation followed by a pooling, the extracted features are constrained to have non-negative entries, hence cannot satisfy an MSC arrangement. Therefore, we add a fully connected layer with the same input and output dimensions[5] on top of ResNets as the feature extractor $f()$, such that the output feature is theoretically defined over $\mathbb{R}^m$. The choice of hyper-parameters $\tau^+$ and $\tau^-$, as well as additional analysis, are discussed in Section 5.

## 4.1 FINE-GRAINED CLASSIFICATION

**Experimental Setup.** For this setup, we follow Kornblith et al. (2021) and first pre-train a ResNet-50 model (He et al., 2016) on ImageNet ILSVRC 2012 dataset (Russakovsky et al., 2015). The outputs of ResNet-50 are then used as features for the downstream tasks. More specifically, after pre-training, we fix the pre-trained feature extractor and only learn a linear classifier for each of the nine different downstream natural image classification tasks through $L_2$-regularized multinomial logistic regression. The downstream datasets are Food (Bossard et al., 2014), CIFAR10 & CIFAR100 (Krizhevsky et al., 2009), Birdsnap (Berg et al., 2014), Sun (Xiao et al., 2010), Car (Krause et al., 2013), Aircraft (Maji et al., 2013), Pet (Parkhi et al., 2012) and Flower (Nilsback & Zisserman, 2008). Most of these datasets concern recognition on fine-grained classes (see Appendix A for more details). The primary goal of this experiment is to demonstrate that *diverse features* in pre-training enables better downstream fine-grained classification.

**Results.** The results for fine-grained classification are summarized in Table 1. We consider several baselines to evaluate our performance gain. We use CE to denote the baseline which jointly learns the classifier and the feature extractor from scratch by minimizing CE loss $\mathcal{L}_{CE}$ during pre-training. Another natural baseline is CE+ETF, where the learnable classifier in CE is replaced with a fixed Simplex ETF. To have a fair comparison with our HingeCE which also uses fixed Simplex ETF as the classifier but uses an additional linear layer, we also add such a layer for CE+ETF. Hence, HingeCE and CE+ETF are the same in terms of architecture of feature extractor and using fixed Simplex ETF classifier, with the only difference being the choice of loss function for network training.

Additionally, following Kornblith et al. (2021), we use label-smoothed variants of the baselines above as additional baselines (e.g. Smooth+ETF). In particular, we modify the target distribution $t$ as follows $t' = t(1-\alpha)+\alpha/K$ where $\alpha$ determines the weighting of the original and uniform targets and $K$ is the number of classes in Eq. 7. We set $\alpha$ as 0.1 and mean linear evaluation accuracy drops from 94.3% to 92.6% (Row$_{(2,3)}$ of Table 1) when a classifier is jointly learned and from 94.0% to 92.7% (Row$_{(4,5)}$ of Table 1) when a Simplex ETF is used. We first note that CE+ETF achieves similar or competitive accuracy in linear evaluation as well as the accuracy on ImageNet (pre-training) test set when compared to CE (Row$_{(1,2)}$ of Table 3 ), indicating that using Simplex ETF as classifier does not impact the transferability of deep learning features. Finally, with our approach HingeCE, we mitigate the degree of collapse by maintaining diversity within each class, and observe substantial performance boost over the baselines on both transfer learning (Row$_{(5,6)}$ of Table 1) and test accuracy on ImageNet (Row$_{(2,3)}$ of Table 3). Also, in contrast to self-supervision approaches (e.g.

---

[5]For CIFAR100-LT classification using ResNet-32, the number of class $K = 100$ is greater than the feature dimension $m = 64$ which violates the constraint of $m \geq K$ for constructing a Simplex ETF in (3). In this particular case we set the output dimension to be 128.

Table 2: Performance comparison on long-tail datasets. We report Top-1 accuracy (%) and the **best** accuracy is highlighted. Our implementation is based upon Menon et al. (2020). Symbol $^*$: the accuracy on is 38.4% when the feature dimension $m = 64$ and is 39.5% when $m = 128$. To ensure Simplex ETF constraint holds, for CIFAR100-LT, we increase the feature dimension to 128.

| Approach | CIFAR10-LT | CIFAR100-LT | ImageNet-LT |
|---|---|---|---|
| Focal (Lin et al., 2017) | 70.4 | 38.4 | - |
| LADM (Cao et al., 2019) | 73.4 | 39.6 | - |
| M2M (Kim et al., 2020) | 79.1 | 43.5 | - |
| OLTR (Liu et al., 2019) | - | - | 32.2 |
| cRT (Kang et al., 2019) | - | - | 47.3 |
| FCL (Kang et al., 2020) | - | - | 49.8 |
| KCL (Kang et al., 2020) | 77.6 | 42.8 | 51.5 |
| TSC (Li et al., 2022) | 79.7 | 43.8 | **52.4** |
| Logit Adjusted (Menon et al., 2020) | 77.7 | 43.9 | 51.1 |
| CE (Menon et al., 2020) | 72.8 | 38.4* | 46.9 |
| CE+ETF | 77.3 | 42.6 | 48.1 |
| HingeCE (Ours) | **80.4** (↑ 2.9) | **46.2** (↑ 3.6) | 50.8 (↑ 2.7) |

SimCLR) which aim to learn diverse features, our method can provide both discriminative power and feature diversity, thereby, yielding better transfer learning performance.

## 4.2 LONG-TAIL CLASSIFICATION

**Experimental Setup.** For long-tail classification, we conduct experiments on three popular long-tail benchmarks: CIFAR10-LT, CIFAR100-LT (Krizhevsky et al., 2009), and ImageNet-LT (Russakovsky et al., 2015). CIFAR10-LT and CIFAR100-LT are derived from CIFAR10 and CIFAR100 with the number of samples per class ranges from 5000 to 50 and 500 to 5, respectively. ImageNet-LT is derived from ImageNet and contains 1000 classes. The number of samples per class ranges from 1280 to 5. Based on the number of training samples per class, we group the classes into two categories: *major class* which have more than 50 samples, and *minor classes* with less than 50 samples. In our method, we tune a $\tau^+$ for the minor classes and simply use $\tau^+ = 0$ (which is equivalent to using $\langle w_k, u_i \rangle$ as logits) for the major classes.

**Results.** In addition to standard baselines, we also compare our approach with state-of-the-art long-tail classification methods. The results are summarized in Table 2. Using Simplex ETF as classifier, CE+ETF already yields improved performance when compared to CE (Row$_{(10,11)}$ of Table 2) as class centers are predefined and maximally separated while the centers of minor classes may not be well-learned (Fang et al., 2021). For CIFAR100-LT, even though the test accuracy improved from 38.4% to 39.5% by increasing feature dimension from 64 to 128, a larger gain in test accuracy (from 39.5% to 42.6%) was obtained through the use of Simplex ETF. Note that the performance gain using CE+ETF on ImageNet-LT is relatively minor compared to CIFAR-related datasets. On the other hand, by preserving the diversity of each class, our HingeCE considerably improves the performance on all three datasets yielding state-of-the-art performance on a few datasets and is competitive on the rest. Note that TSC (Li et al., 2022) and KCL (Kang et al., 2020) use two-stage training involving supervised contrastive learning (SCL) to improve the performance on long-tail classification, which is complementary to our approach. Nonetheless, even without SCL, we obtain better or competitive results. We believe our results can be further improved using SCL and additional hyperparameter tuning.

## 5 DISCUSSION

**Choice of Hyper-parameters.** $\tau^+$ and $\tau^-$ are two hyper-parameters of our method introduced in Equation 8. Here we plot the cosine similarities between each feature on ImageNet train set with its positive class center and rest of the negative class centers during the training process, and present the results in Figure 3(a) and (b), respectively. It can be observed that the mean cosine similarity with positive class centers is around 0.46, which is perhaps larger than expected (since if all features collapse completely to class mean, then this value should be one). Moreover, the similarity with

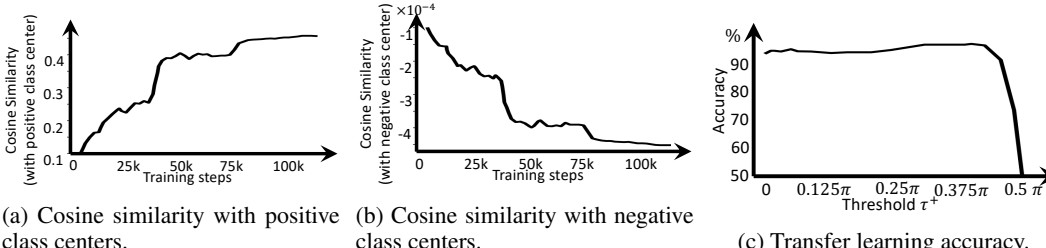

(a) Cosine similarity with positive class centers.

(b) Cosine similarity with negative class centers.

(c) Transfer learning accuracy.

Figure 3: (a, b) Cosine similarity of each feature with its corresponding positive (in (a)) and negative (in (b)) class centers averaged over all features obtained with CE+ETF. (c) Transfer learning accuracy on Flower dataset as a function of $\tau^+ \in [0, \frac{17}{36}\pi]$ with $\tau^- = \pi$ during ImageNet pretraining with HingeCE.

negative class centers remains close to zero during the entire training process. Hence, the negative logits have small impact on the value of the CE loss and for simplicity we set $\tau^- = \pi$ (i.e., use $\langle \boldsymbol{w}_k, \boldsymbol{u}_i \rangle$ as logits). In the following we examine the effect of $\tau^+$.

Figure 3(c) demonstrates the effect of $\tau^+$ for fine-grained classification on Flower dataset. It can be seen that with a small value of $\tau^+$, HingeCE does not have a significant effect. This is aligned with the observation in Figure 3(a). When $\tau^+$ is increased to the range of $\frac{11}{36}\pi \leq \tau^+ \leq \frac{5}{12}\pi$, the accuracy increases considerably and is not sensitive to the choice of $\tau^+$ within that range.

For all results in Table 1 & 3, the value of $\tau^+$ is fixed to $\frac{13}{36}\pi$. Since this choice works fairly well across various settings, hyper-parameter tuning for our method is very easy. Finally, while this fixed choice works well, we find empirically without reporting the details here that additional hyper-parameter tuning can further improve the final performance.

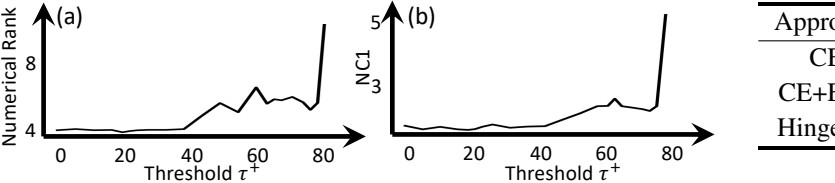

| Approach | Accuracy (%) |
|----------|--------------|
| CE | 76.5 |
| CE+ETF | 76.3 |
| HingeCE | 77.4 |

Figure 4: The (a) numerical rank and (b) NC1 metric among features on the ImageNet training set with different thresholds $\tau^+$ for $h^+$.

Table 3: Top-1 test accuracy on ImageNet.

**Measuring Diversity.** To measure the diversity of features, we use $\mathcal{NC}1$, defined through between-class and within-class covariance matrices, and numerical rank, defined as the ratio of squared nuclear norm and squared Frobenius norm for features of each class averaged over all classes. The definition of these two measures can be found in e.g., Zhou et al. (2022). In Figure 4, we plot $\mathcal{NC}_1$ and numerical rank as a function of $\tau^+$. It can be seen that both measures increases with $\tau^+$, showing that our method indeed improves feature diversity.

**Pre-training Performance.** While designed for improving the performance of downstream fine-grained classification task, we show in Table 3 that our method improves the performance on the test set of ImageNet as well.

**Per-class Analysis for Long-tail Classification.** We evaluate the performance gain of our method on major vs minor classes in long-tail classification, and report the results in Figure 5. Specifically, Figure 5(a) & (b) shows that the benefit of our method is mainly on the minor classes. Figure 5(c) demonstrates the effect of $\tau^+$ on major and minor classes separately. It can be seen that setting $\tau^+$ to be in the range of $\frac{7}{36}\pi \leq \tau^+ \leq \frac{1}{4}\pi$ provides a notable gain for minor classes. In addition, setting $\tau^+$ to be in the range of $\frac{1}{12}\pi \leq \tau^+ \leq \frac{11}{90}\pi$ (which is a range with lower values than that of the minor classes) provides a gain for major classes. This may imply that the optimal threshold $\tau^+$ should be dependant upon the number of training samples in each class. Hence, while the results in Table 2 are obtained from using the same $\tau^+$ in all minor classes and another $\tau^+$ for all major classes, which is a choice made to simplify the study, the performance may be improved with a more careful choice of $\tau^+$ for individual classes.

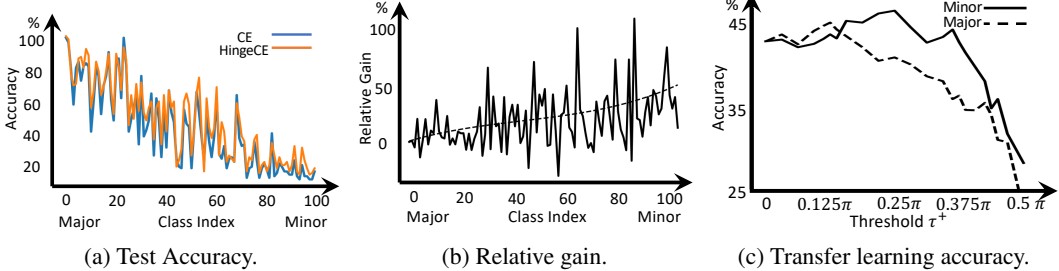

(a) Test Accuracy.        (b) Relative gain.        (c) Transfer learning accuracy.

Figure 5: On CIFAR100-LT, for each class, we calculate (a) test accuracy and (b) relative performance gain (the class with a larger index on x-axis has less training data) and the performance on minor class are mostly improved. Then, we apply our approach on the major and minor classes separately and adjust the threshold $\tau^+$. In (c), we observe that the best testing accuracy is reached when our approach is applied on minor classes and the threshold is around $\frac{1}{6}\pi \leq \tau^+ \leq \frac{5}{18}\pi$.

## 6   RELATED WORK

**Transfer Learning** is a pipeline where feature extractors trained using a pretraining task are used to perform downstream tasks (Kornblith et al., 2019). Existing literature on transfer learning abounds and most related to ours include Kornblith et al. (2021) which show a trade-off between pretraining and downstream performance. Specifically, many techniques such as label smoothing produces more collapsed within-class features (Müller et al., 2019) hence improves the performance on ImageNet pre-training task, but such features lead to worse downstream performance. Our result shows that by introducing diversity, the performance on both pretraining and the downsteam fine-grained classification tasks can be improved. While other methods for obtaining diverse features exist, they either require an additional $k$-nearest neighbor search (Dwibedi et al., 2021; Touvron et al., 2021; Feng et al., 2022), or require that different classes share the same feature distribution (Xu et al., 2021). On the other hand, within-class compactness is found to benefit face recognition (Wen et al., 2016; Liu et al., 2017; Deng et al., 2019; Wang et al., 2018; Meng et al., 2021).

**Neural Collapse** is a phenomenon reported in Papyan et al. (2020) for deep models fully trained on a balanced dataset in classification tasks. It simplifies the deep model by considering the last-layer features and the classifier as independent variables, and subsequent works theoretically demonstrated phenomenon of $\mathcal{NC}$ under Cross-Entropy (Wojtowytsch & Weinan, 2020; Lu & Steinerberger, 2020; Graf et al., 2021; Fang et al., 2021) or Mean Squared Error (Mixon et al., 2022; Rangamani & Banburski-Fahey, 2022; Poggio & Liao, 2020) losses. Since the classifier weight forms a Simplex ETF in $\mathcal{NC}$, Zhu et al. (2021) shows that fixing the classifier as a Simplex ETF during training can achieve similar performance. While the minimized diversity may hurt the adaptation performance to new classes (Kornblith et al., 2019), the Simplex ETF has been shown to be useful for continual learning (Wu et al., 2021) and long-tail learning (Li et al., 2022; Yang et al., 2022).

**Long-tail Learning** with imbalance of data distribution is a common real-world scenario. Deep learning models tend to perform well on major classes (Zhang et al., 2021) with a lower accuracy on minor classes. To address the issue, an intuitive strategy is to explicitly re-sample the training data (Ando & Huang, 2017) which usually hurts performance on major classes (Kang et al., 2019). Alternative methods include re-weighting the loss for different classes Byrd & Lipton (2019); Cui et al. (2019) or re-adjusting the logits (Menon et al., 2020; Cao et al., 2019) during training. Meanwhile, several regularization techniques such as weight-normalization (Kang et al., 2019) and loss modification (Iranmehr et al., 2019) can be applied, but they can be sensitive to the selection of optimizer and may not reach a balanced error (Menon et al., 2020).

## 7   CONCLUSION

In this paper, we ask the question of whether Neural Collapse is a desirable property of deep features and argue that the lack of within-class diversity may hurt the performance on fine-grained classification task. To address that, we introduce a geometric arrangement of features with within-class diversity, and present a method called HingeCE based on the classifier and loss function design. Empirically we show that HingeCE is substantially better than or competitive to the state-of-the-art methods for fine-grained classification and long-tail classification tasks.

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

## A    FINGRAINED TRANSFER DATASET

The downstream datasets are Food (Food-101) (Bossard et al., 2014) with 101 finegrained classes of food, CIFAR10 & CIFAR100 (Krizhevsky et al., 2009), Birdsnap (Berg et al., 2014) with 500 finegrained classes regarding bird species, Sun (SUN-397) (Xiao et al., 2010) with 397 scene categories, Car (Stanford Cars) (Krause et al., 2013) with 196 finegrained class of car models, Aircraft (Maji et al., 2013) with 120 finegrained classes of aircraft models, Pet (Oxford IIIT Pets) (Parkhi et al., 2012) with 37 finegrained classes of dog & cat, and Flower (Oxford IIIT Flowers) (Nilsback & Zisserman, 2008) with 102 finegrained classes regarding flower species. Food(Bossard et al., 2014) contains 101 finegrained classes For all datasets, we follow Kornblith et al. (2021) and use the official train-test split for linear evaluation training. Then, for evaluation, on the test set, we report the top-1 accuracy for Food, CIFAR10, CIFAR100, Birdsnap, SUN and Cars; and mean per-class accuracy for Aircraft, Pets, Flower.

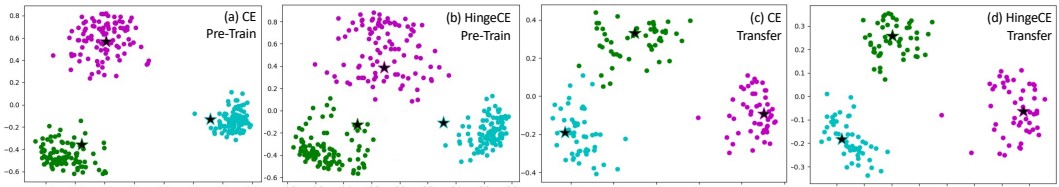

Figure 6: Visualization comparison ImageNet (pre-train) and Flower (Transfer learning). Comparing with baseline CE, by applying HingeCE, as the features are more diverse on pre-train dataset ((b) vs. (a)), the features for finegrained classes more separable ((d) vs. (c)), which then facilitate the finegrained recognition in transfer learning.

**Visualization on fingrained classes.**    As shown in Figure 6, we provide a visualization on three coarse classes for ImageNet and three Flower separately. All of the classes are randomly selected. Comparing Figure 6(a,b), applying our approach HingeCE can preserve more diversity within each coarse class. Then, from Figure 6(c,d), the features of finegrained classes can be more separable.

## B  IMPLEMENTATION DETAILS

We use the ResNet for all experiments. For each dataset, we follow the same configuration of ResNet used in the baseline approach for fair comparison. Meanwhile, as ResNet has a ReLU activation before average pooling He et al. (2016), the feature vector (after pooling) cannot satisfy the constraint of the defined Simplex ETF in an MSC arrangement, *i.e.*, the weights in simplex ETF are maximally separated over the full feature space (always have negative entries) while the averaged feature vectors always have non-negative entries and its distance with with the weights then cannot be zero. As such, we add a fully connected layer, serving as a projector, on top of the resnet structure such that the value of each entry in the output features is theoretically defined over $\mathbb{R}^m$

As the thresholds $(\tau^+, \tau^-)$ are the only hyper-parameters introduced in our approach where $\tau^+ = \tau$ and $\tau^-$ is usually set as 0, we summarize them in the Table 4 and explain the details of other hyper-parameters below.

Table 4: Summary of threshold selection

| Dataset | $\tau^-$ | $\tau^+$ | Range | The classes on which HingeCE are applied. |
|---|---|---|---|---|
| ImageNet | | $\frac{13}{36}\pi$ | $\frac{11}{36}\pi \leq \tau^+ \leq \frac{5}{12}\pi$ | All 1000 classes |
| ImageNet-LT | 0 | $\frac{5}{18}\pi$ | $\frac{2}{9}\pi \leq \tau^+ \leq \frac{11}{36}\pi$ | 383 minor class |
| CIFAR10-LT | | $\frac{1}{3}\pi$ | $\frac{5}{18}\pi \leq \tau^+ \leq \frac{5}{12}\pi$ | 1 minor class |
| CIFAR100-LT | | $\frac{2}{9}\pi$ | $\frac{7}{36}\pi \leq \tau^+ \leq \frac{1}{4}\pi$ | 50 minor class |

The class with $\leq 50$ training data are treated as minor classes.

### B.1  PRETRAINING ON IMAGENET

We use the standard Resnet-50 as feature extractor and follow the configuration defined in He et al. (2016). For the projector, we set the output dimension the same as the input dimension 2048. We set the batch size as 512, use SGD optimizer with nesterov, and train for 90 epoches. The coefficient of weight decay is 0.001. We set the initial learning rate as 0.4 with a linear warm-up of 100 steps and scheduled according to a cosine annealing function.

### B.2  TRANSFER LEARNING FOR FINE-GRAINED CLASSIFICATION

During transfer learning, we fix the feature extractor learned by our HingeCE to extract the 2048-dim feature vector for each image. Then, we learn a linear classifier on top of the extracted features from images of train set. As the dataset considered in transfer learning are abount fine-grained classification, we use the performance on fine-grained classification to indicate the quality & diversity of the learned features.

Note that the classifiers in transfer learning are trained by minimizing vanilla cross-entropy loss and we do not apply our approach in transfer learning. During training, we set the batch size as 128, use SGD optimizer, and train the classifier for 100 epoches. As only a linear classifier is learned, we set a large learning rate 1.0 in the whole training process. The information of dataset is summarized in Section A.

### B.3  LONG-TAIL LEARNING

We use ResNet-32 as backbone for experiments on CIFAR10-LT and CIFAR100-LT, and ResNet-50 for experiments on ImageNet-LT. For each dataset, all of the approaches being compared use the same architecture as ours and the comparison is fair regarding network architecture. The implementation details for each dataset can be found below.

**CIFAR10-LT**  We use the standard ResNet32 architecture in our experiments and exactly follow the configuration in Menon et al. (2020); Yang & Xu (2020). As the spatial resolution of image is $32 \times 32$, the first convolution layer is with stride 1 and a kernel of size 3x3. The number of channels

in the feature output is 64 and the projector linear layer has the same input and output dimension (*i.e.*, 64). We set the batch size as 128, use SGD optimizer, and train for 256 epoches. We set the coefficient of weight decay as 0.0001. We set the initial learning rate as 0.4 without any warm-up and scheduled according to a cosine annealing function.

**CIFAR100-LT**  We use the same network (*i.e.*, ResNet32) as feature backbone and follow the experiment configuration described in CIFAR10-LT except for the configuration of projector. As the number of classes in CIFAR100-LT is $K = 100$ and larger than the feature dimension $m = 64$, we thus set the output dimension $m = 128$ to meet the constraint have $m \geq K$. To note, the structure of ResNet32 is not changed.

**ImageNet-LT**  We use the same network (*i.e.*, ResNet50) as feature backbone and experiment setup described in Section B.1 but we use 0.0001 as the coefficient of weight decay.

