# OpenReview forum: "Do We Need Neural Collapse? Learning Diverse Features for Fine-grained and Long-tail Classification"
_ICLR.cc/2023/Conference — Submitted to ICLR 2023_

### Official Review · Reviewer_ur4C · 2022-10-26

**Confidence:** 3
**Correctness:** 2
**Technical Novelty And Significance:** 2
**Empirical Novelty And Significance:** 2
**Recommendation:** 5

**Clarity, Quality, Novelty And Reproducibility:**

Clarity: Good
Quality: Good
Novelty: Generally a new technique but it's not a surprising one. Similar ideas have been proposed for faces [mag face, arc Face]. I would like the authors to discuss them and clarify their novelty.
Reproducibility: Good.

**Strength And Weaknesses:**

Strengths:
- The paper presents a simple idea and is easy to follow.
- The method is appropriately designed following the main idea.

Weaknesses:
- My main concern here is that the paper doesn't offer an explanation of why NC is undesirable for fine-grained classification, i.e., the main idea isn't well justified. I think NC should be characterized carefully in order to support the main idea.  In particular, I disagree that "collapsing to the class mean" is a good enough reason to say NC is undesirable for fine-grained classification. The authors can convince me by quantifying the collapse and demonstrating how it correlates with the fine-grained classification performance. As of now, I need to believe that it's the case and without any proof of concept. Personally, I think "collapsing to the class mean" is quite vague and unclear. For example, one possible interpretation is that the distances between features of the same class are small w.r.t the distance between classes. In this case, it doesn't mean that the feature doesn't encode enough fine-grained information for each instance since intra-class variance can still be high.

- Likewise, since the method is based on a weak motivation (to me), I am not sure how rearranging features into a cone helps with fine-grained classification. There are methods that propose feature space re-arrangement before (Mag-face, arc-face) and they are very clear about why their geometrical constraints boost their performance. I don't see a similar explanation here. Is there any relation between the proposed  MSC and intra-class variance [1]? If there is a connection here that would add depth to the paper and offer an explanation for what is happening.
[1] Variational Feature Disentangling for Fine-Grained Few-Shot Classification - ICCV21

- The paper introduces some hyperparameters that need to be tuned.

- The paper doesn't improve the performance on the most challenging case of the imagenet-LT.

**Summary Of The Paper:**

The paper discusses NC - intra-class features collapse to the class mean and different class means are maximally separated. The authors argue that NC is an undesirable property for fine-grained classification. Thus they constrain the features of the same class should lie in a cone instead of collapsing to the mean. A feature extractor pre-trained by this method improves fine-grained classification on multiple datasets.

**Summary Of The Review:**

Overall, I am on the fence. The paper is simple and the method seems to work in some cases. The numbers are not substantially strong. The explanation and motivation aren't clear. I am not an expert in this area so please fell free to correct me.

---

> ### Author Response · Authors · 2022-11-16
> **Response to Reviewer ur4C (1/2)**
>
>
> **Question 1**: The paper doesn't offer an explanation of why $\mathcal{NC}$ is undesirable for fine-grained classification.
>
> A1: We provide clarification on the notion of "collapsing to the class mean" in $\mathcal{NC}$, which we believe can fully address this concern.
>
> First, we emphasize that theoretically, "collapsing to the class mean" means that the class variance becomes *exactly* zero, i.e., all features of a particular class are identical. Specifically, it can be proved (see Section 6 for a short review) under certain simplifying assumptions, e.g., the unconstrained feature model, that all global solutions have zero class variance. It is easy to see that when class variance becomes exactly zero, such features cannot be used for downstream fine-grained classification. This is what we meant by saying that $\mathcal{NC}$ is undesirable for fine-grained classification.
>
> Second, practical neural network training can not expect the variance to be *exactly* zero, both because the assumption of the unconstrained feature model is not fully satisfied and because the training is terminated at finite steps. However, this does not mean we should not study whether the collapse in practice already hurts transfer ability. In other words, the theoretical results for $\mathcal{NC}$ mentioned above provide sufficient motivation for us to take a close look at the effect of collapsing and diversity in practice. As it turns out, our empirical results in Figure 3 (c) show that allowing features to be diverse indeed helps to improve transfer learning accuracy with fine-grained classes, which justifies the value of our study.
>
> Specifically, for the example shown in Figure 1, we train a classifier on CIFAR-20. For the baseline CE, the $l_2$ distance between features and the ground truth centers is $0.64 \pm 0.19$, the numerical rank and $\mathcal{NC}1$ are 2.97 and 0.13 separately. In contrast, by our HingeCE, we can increase the distance to $9.28 \pm 2.42$, the numerical rank and $\mathcal{NC}1$are 4.51 and 1.37. This comparison quantitatively demonstrates that "collapsing to the class mean" will minimize the class variance and cannot preserve enough fine-grained information.
>
> **Question 2**: Is there any relation between the proposed MSC and intra-class variance [1]?
>
> A2: While both MSC and [1] aim to learn diverse features, the method in [1] imposes that the within-class distribution is the same for all classes. In contrast, MSC does not have this constraint as each class may have a different distribution within its corresponding cone. We added a short discussion in section 6.
>
> **Question 3**: Hyper-parameter tuning.
>
> A3: Like all other hyper-parameters in training a deep neural network, $\tau$ can simply be chosen via cross-validation. We note that $\tau$ is the only hyper-parameter introduced by our method and Figure 3 shows that the performance is not sensitive to the choice of $\tau$. Hence our method is easy to use.
> For longtail learning, while our current strategy is to use the same $\tau$ for all minor classes, our method may further benefit from using different values of $\tau$ for different classes. We note that this may be practical by choosing $\tau$ based on the class probability (a similar idea was successfully used in prior work such as [Menon et al., 2020]). We leave an extensive study on this idea to future work.
>
> **Question 4**: The paper doesn't improve the performance on the most challenging case of the imagenet-LT.
>
> A4: This is not true. Our method improves the performance on ImageNet-LT from 46.9\% to 50.8\%.
>
> Perhaps the reviewer meant that our method is not as good as some other competing methods. As discussed at the end of section 4, the TSC (Li et al., 2022) and KCL (Kang et al., 2020) use two-stage training involving supervised contrastive learning (SCL) to improve the performance of longtail classification.
>
> Meanwhile, please note that the implementation is not the same, and comparing HingeCE with them is not fair. For example, KCL employs both color jittering and flipping as data augmentation for training, but we only use image flipping. To understand the effect of jittering, we used the official KCL GitHub repo and obtained its performance as 51.2%. Then, we remove the jittering for KCL during training and its performance drops to 49.6%, which is lower than our HingeCE (50.8%). Hence, under a fair comparison, our method is also better than KCL on ImageNet-LT.
>
> In addition, as discussed in TSC, SCL has the advantage of preserving the neighbor uniformity for features in the same class, which could benefit long-tail classification. However, the focus of our MSC features is on the diversity of features under the context of neural collapse. Hence, the comparison to them is not fair, and we believe that our results can be further improved by similarly introducing SCL to our approach as TSC and KCL.

---

> > ### Author Response · Authors · 2022-11-16
> > **Response to Reviewer ur4C (2/2)**
> >
> >
> > **Question 5**: Similar ideas have been proposed for faces [mag face, arc Face].
> >
> > A5: ArcFace, as well as similar methods such as large-margin softmax [a], angular softmax [b], and large-margin cosine loss [c], are based on the idea of increasing the discriminative power of features while enforcing within-class compactness. Their methods do not aim to encourage diversity of features within each class, so they are very different from ours.
> >
> > MagFace aims to encode confidence in the magnitude of the features, and by encouraging the magnitude to be large for simple examples, it reduces overfitting to hard samples. That said, it is not designed for the features to become more diverse within each class, hence is fundamentally different from our approach. We added a citation to these relevant works in section 6.
> >
> > [a] Weiyang Liu, Yandong Wen, Zhiding Yu, and Meng Yang. Large-margin softmax loss for convolutional
> > neural networks. ICML 2016
> >
> > [b] Weiyang Liu, Yandong Wen, Zhiding Yu, Ming Li, Bhiksha Raj, and Le Song. Sphereface: Deep
> > hypersphere embedding for face recognition. CVPR 17
> >
> > [c] Hao Wang, Yitong Wang, Zheng Zhou, Xing Ji, Dihong Gong, Jingchao Zhou, Zhifeng Li, and Wei
> > Liu. Cosface: Large margin cosine loss for deep face recognition. CVPR 18

---

> > ### Comment · Reviewer_ur4C · 2022-12-07
> > **Thank you for your reponse.**
> >
> > > "collapsing to the class mean" means that the class variance becomes exactly zero
> >
> > Thank you for this clarification. I now understand that what we do here is to relax the classification loss such that the class features do not collapse to the same point, i.e.,  the intra-class variance is zero. I think the high-level concept is widely used before but without NC mentioned and thus, I am afraid that we are reinventing the wheels here. I am not sure should we even need to discuss NC in the introduction in order to fully understand the contribution of the paper. I personally find it unnecessary and somewhat distracting. Second, if we break down the core concept of the paper into preventing the features to collapse to the class mean (or making them diverse), then there should be a long discussion about how previous methods achieve this goal (or similar goal) including many effective long-tailed losses:
> >
> > - Arcface Loss
> > - Seesaw Loss - CVPR 21
> > - Polyloss - Polyloss: A polynomial expansion perspective of classification loss functions - ICLR 22
> > - Circle Loss - CVPR 22
> > - Additive Margin Softmax Loss - Signal Processing Letters 25 (2018)
> >
> > Please note that here I am not asking for specific differences between the proposed loss and these losses. I am mostly concerned about missing a proper high-level view of the goal that we are trying to achieve, which leads to missing potential crucial discussions. To me, fixing "collapsing to the class mean" can be obtained by almost any losses using a soft margin. Why is arranging features into cones (with the same angular radius) superior to all previous losses? What happens if I use a simple loss function that stops the gradient when the feature is "close enough" to the class mean? It's hard for me to grab the importance and significance of the proposed loss function.
> >
> > sub-question: does the angular radius have to be the same across classes?
> >
> > > "Their methods do not aim to encourage diversity of features within each class, so they are very different from ours."
> >
> > What particular thing in the proposed method that enforces diversity?

---

> > > ### Author Response · Authors · 2022-12-12
> > > **New Response (1/2)**
> > >
> > > We thank the reviewer for the response.
> > >
> > > >  the high-level concept (i.e., the class features do not collapse to the same point) is widely used before.
> > >
> > > It is not clear to us what methods the reviewer is having in mind by saying that it is "widely used before"
> > >
> > > If by "widely used" the reviewer refers to the work of Arcface, Seesaw, Polyloss, Circle Loss, and Additive Margin Softmax Loss, we mention that these methods are not designed to avoid collapsing of within-class features, and none of them provide evidence that their losses produce diverse features. In fact, Polyloss is a generalization of CE and Focal losses, both of which enforce collapsing rather than avoid it. Arcface is designed to "maximize class separability", and Additive Margin Softmax Loss is designed to explicitly minimize intra-class variation, which have an unknown effect for within-class diversity.
> > >
> > > If by "widely used" the reviewer refers to the work such as [a, b] based on k-nearest neighbor search, we note that those works are considerably more complicated and compute-demanding than our method. We have added a discussion with more related work to Section 6. Furthermore, as discussed in Section 5, we use $\mathcal{NC}$1 and numerical rank to quantitatively represent the feature distribution, and also provide ablation study to discuss the correlation between choice of threshold and the feature diversity. In this way, it is very clear what features we are learning.
> > >
> > > [a] Rethinking supervised pre-training for better downstream transferring. ICLR 2022
> > >
> > > [b] Grafit: Learning fine-grained image representations with coarse labels. CVPR 2021
> > >
> > > > I am not sure should we even need to discuss NC in the introduction in order to fully understand the contribution of the paper.
> > >
> > > NC is a critical part of the paper firstly because it provides a motivation of the problem. Specifically, we agree that one can argue for the benefit of diverse features without mentioning NC. However, this does not serve as sufficient motivation for designing approaches for obtaining diversity, for the reason that, it is not known whether lack of diversity is a property of features learned via standard method at all. NC provides a strong motivation because it clearly states that within-class features are indeed collapsing.
> > >
> > > In addition, NC also enabled us to design our method. Specifically, NC provides a mathematical understanding on the geometric arrangement of deep features. Our approach is based on modifying that arrangement by directly hand-crafting the desired geometric arrangement of the features (i.e., the maximal-separating-cones).

---

> > > > ### Author Response · Authors · 2022-12-12
> > > > **New Response (2/2)**
> > > >
> > > > > missing a proper high-level view of the goal that we are trying to achieve.
> > > >
> > > > At a high-level, the goal of this paper is to provide a new perspective for obtaining features with desired properties, based on directly handcrafting the geometric arrangement of the features.
> > > >
> > > > Specifically, there are a ton of existing methods for obtaining features with desired properties, such as those based on modifying the gradient, regularizing the features, and loss design, for various purposes, e.g., long-tail learning. While such methods have achieved great success in obtaining features with desired property, they work in a "black-box" fashion, namely, it is unclear what the geometric arrangement of the so-learned features is. This is largely due to the fact that there has not been a clear understanding on the deep features learned from standard training pipeline.
> > > >
> > > > Our method is motivated by the understanding provided by NC to directly handcraft the geometric arrangement of features. As such, our method is a "white-box" method, where e.g., the hyper-parameter of the method is directly connected to the angular radius of the pre-set cones. In this paper we focused on the case of within-class diversity to illustrate our methodology, with the hope that this new perspective may help to design generalizable solution to different tasks (e.g., the fine-grained transfer learning and long-tailed learning evaluated in this paper) and  appropriate geometry of features for other tasks as well.
> > > >
> > > > > What happens if I use a simple loss function that stops the gradient when the feature is "close enough" to the class mean? Why is arranging features into cones is better?
> > > >
> > > > One can indeed do what the reviewer suggested to stop the gradient, and we believe this will give rise to features with desired within-class diversity. In fact, part of our method is based precisely on this idea of stopping the gradient when the features are close to class mean.
> > > >
> > > > However, such an approach on its own has the drawback that, it is unknown *if the learned class centers are sufficiently separated*. This is precisely the drawback of such an approach based on solely enforcing certain properties of the features without really understanding the geometry of the learned features. In contrast, in our approach, we use the weights in Simplex ETF as anchor for class centers, and the selection of Simplex ETF is based on the understanding in NC.
> > > > Then, for each anchor defined by Simplex ETF, we pre-set a cone such that all cones are also maximally separated from each other. In this way, we can expect the features to achieve the desired inter-class separation and intra-class diversity at the same time.
> > > >
> > > >
> > > > > does the angular radius have to be the same across classes?
> > > >
> > > > Not necessary. We used the same angular radius for all classes in our experiment for simplicity. One can certainly explore using separate radius for different classes, which we leave for future work.
> > > >
> > > >
> > > > > What particular thing in the proposed method that enforces diversity?
> > > >
> > > > Our method enables us to learn diverse features by stopping the gradient for data points that are already close enough to its class mean. As explained in Theorem 1, this enables the network to stop updating once the features are in a MSC configuration.

---

> ### Author Response · Authors · 2022-12-07
> **Further discussion**
>
> We would like to thank the reviewer again for providing the detailed comments for our paper.
>
> We have provided a detailed response to all the raised questions on Nov 16 and added new results on ImageNet-LT to our response of question 4 recently. As the discussion period is ending soon, we would like to check back and see if there are any remaining questions or concerns. We'd be happy to engage in further discussions with the reviewer.

---

### Official Review · Reviewer_6L41 · 2022-10-28

**Confidence:** 4
**Correctness:** 3
**Technical Novelty And Significance:** 3
**Empirical Novelty And Significance:** 3
**Recommendation:** 8

**Clarity, Quality, Novelty And Reproducibility:**

The paper is clearly written and technically sound, which proposed a novel feature property named maximal-separating-cone.

**Strength And Weaknesses:**

Pros:

+ Well-motivated approach: Investigates an intriguing phenomenon of NC, and its undesirable properties for fine-grained classification adaptation.
+ Technically sound solutions with a novel hinge function design.

Cons:
- It is unclear to me why learning MSC features can also benefit long-tailed classification tasks. Experimental results in Table 2 also seem to show weaker performance gain on long-tailed datasets.


**Summary Of The Paper:**

This paper studies the undesirable Neural Collapse (NC) phenomenon for specific classification tasks, including fine-grained classification and long-tailed classification. The authors then propose to learn deep features that have the maximal-separating-cone (MSC) property by optimizing towards their designed hinge loss function. Empirical results verified their motivation and the effects of the proposed approach.

**Summary Of The Review:**

This paper is decent research towards learning deep features with MSC property, and hence tackles certain specific classification problems where NC is undesirable.

---

> ### Author Response · Authors · 2022-11-16
> **Response to Reviewer 6L41**
>
> We would like to thank the reviewer for praising our work as well-motivated, technically sound, and clearly written.
>
> The reviewer asks about why MSC features benefit long-tail learning, for which we share our thoughts below.
>
> First, it is previously known (e.g., Fang et al. 2021) that when the classes are imbalanced, the classes are not equally separated in the feature space. Specifically, the distance between minor classes may become much smaller than that between major classes, which hurts the performance of minor classes. MSC feature addresses this issue because all class centers are equally apart from each other, which is achieved in our method by fixing the classifier weight as a simplex ETF. As seen in Table 2, this (i.e., the method CE+ETF) already brings a huge performance gain compared to CE.
>
> Second, comparing CE+ETF and HingeCE in Table 2, both of which fix the classifier as simplex ETF, with the only difference being that HingeCE allows diverse representation, we see that HingeCE provides better performance.
> We conjecture that this is because diverse features are less prone to overfitting, in particular, to the limited training data of minor classes, and allow the model to produce less confident predictions during training. As compared in Figure 5, the relative gain of test accuracy on minor classes is larger. We leave a dedicated study of this conjecture to future work.

---

### Official Review · Reviewer_tRGD · 2022-10-28

**Confidence:** 3
**Correctness:** 2
**Technical Novelty And Significance:** 2
**Empirical Novelty And Significance:** 2
**Recommendation:** 5

**Clarity, Quality, Novelty And Reproducibility:**

The paper is generally written in a very confusing way; the "core" of the method contains unclear parts while it is unclear what Theorem 1 offers.The analysys of NC comes from Papyan et al., 2020 in 2.2, and I dont see how this paper "examines the Neural Collapse phenomenon" as stated in the conclusion. There are also implementation details that are missing in many places and therefore reproducibility is questionable.

### Notes

* Paragraph "Hinge Loss" in 3.2 is unclear in many places, while Eq9 is written in a very confusing way.

* the first paragraph of the intro is in practice merely advocating for the use of learned vs handcrafted features for computer vision, yet, this is common practice in the AI classification since AlexNet won the Imagenet challenge. It can therefore be significantly trimmed. Same with 2.1 that merely describes classification with  cross entropy loss, again the default loss

* The question asked in the introduction is ill-posed as expressed there. "the" desired property implies that it would be the only, while "deep features" have a multitude of uses- which use does this paper care about is not mentioned till there in the intro. The authors right after mention how they are only looking at a specific task, that of coarse-to-fine classification. The intro needs revision.




**Strength And Weaknesses:**

### Strengths

* The paper dealing a very interesting property of learning with cross-entropy, that of class collapse

* the paper is  experimenting on interesting tasks like long-tail recognition and coarse-to-fine learning.

### Weaknesses

W1) Novelty of the paper is questionable: There are some missing references and comparisons that are very related to the proposed. [A] refers to the same issue tackled here, class collapse for transfer learning. Feng et al in [A] do in some sense provide an answer to the question asked in the intro and propose a loss that avoids collapse. Grafit from Touvron et al [B] is even closer to the task and formulation of this paper: it presents an approach for learning with coarse labels and the loss proposed in [B] is a kNN loss therefore also avoiding collapse. The authors should not only discuss but also compare to such methods.  Another missing reference that should be discussed and compared to is wrf fixed classifiers is [C].


W2) Experimental validation is lacking in many regards:

*  It is unclear how this supervised learning method affect the training task of ImageNet in 4.1 The authors compare against SimCLR, a self-supervised method, but not supervised methods like [A,D] in Table 1. They also do not report performance on the training task of imagenet (one more column should be added). As [Kornblith et al 2021], [E] and other papers have mentioned, there exists a tradeoff between training and transfer accuracy.  What is the performance of all methods at the end of training?

* For long-tail, the method is not outpuerforming multiple methods of Table 2 on the larger and less toy-scale ImageNet-LT. Also Fig 5c shows that setting the main hyperparameter \tau is far from trivial. Moreover it is unclear what model is used in each dataset and method in Table2 - are all methods using the same backbone?

### References
[A] Feng, Yutong, et al. "Rethinking supervised pre-training for better downstream transferring." ICLR 2022

[B] Touvron, Hugo, et al. "Grafit: Learning fine-grained image representations with coarse labels."CVPR 2021

[C] Hoffer, Elad,et al  "Fix your classifier: the marginal value of training the last weight layer." ICLR 2018.

[D] Khosla, Prannay, et al. "Supervised contrastive learning." NeurIPS 2020

[E] Sariyildiz et al "Improving the Generalization of Supervised Models." arXiv 2022.

**Summary Of The Paper:**

The authors present a method that tries to rectify feature collapse for classification, arguing that this is an undesirable property for the case where coarse labels are used for training, but we are (also) interested in a finer-grained downstream task. They advocate to fix the classifiers and use a hinge loss. They experiment on coarse-to-fine as well as long-tail classification.

**Summary Of The Review:**

The paper has a number of weaknesses detailed above. I encourage the authors to provide answers to the concerns and questions.

---

> ### Author Response · Authors · 2022-11-16
> **Response to Reviewer tRGD (1/2)**
>
> **Question 1**: Novelty is questionable.
>
> A1: We thank the reviewer for bringing to our attention related work. There are indeed existing work on encouraging within-class diversity in learned features, including [A, B] mentioned by the reviewer. However, the methods in [A, B] require a k-nearest neighbor search, which introduce extra computational cost ([B] stores features for entire *1.1 Million* data on ImageNet) and more complicated training pipelines ([A] requires techniques such as momentum encoder and dynamic adjust of K in neighbor search). In contrast, our method only requires slightly modifying the training loss and fixing the last layer classifier, which does not introduce any extra computation and can be easily implemented with a few lines of code.
>
> Our solution is motivated by a recent line of work on Neural Collapse, which provides a mathematical understanding of the geometric properties of the learned features. In particular, neural collapse motivated us to design a new geometric arrangement of features (i.e., the maximal separating cone), which made the subsequent design of our method quite easy and straightforward. In contrast, the geometry of learned features with methods in [A, B] are less clear (e.g., [B] sets a two-layer MLP as a projector and performs a neighbor search on the features output by projector, but uses the input feature of projector, i.e., outputs of Resnet, for transfer learning while the two feature spaces are not exactly the same). We have added a short discussion to section 6.
>
> Finally, while the idea of fixing the classifier has appeared in [C], it only demonstrated the feasibility of using fixed classifier weight but was not for the purpose of obtaining desired properties of features. In detail, [C] uses Hadamard matrix, a subtype of an orthonormal matrix, as the fixed classifier. The weights in [C] are not as maximally separated as Simplex ETF. The performance may depend on the selection of matrix design and hyper-parameters such as temperature. We have added a discussion in footnote 3.
>
> **Question 2**: No comparison with methods like [A, D] in Table 1.
>
> A2: Our method is based on simply modifying the training loss and fixing the classifier upon regular training with cross-entropy loss. Hence, we use cross-entropy loss as a baseline to demonstrate the benefit of our method. By the results in [Kornblith et al 2021] (``Linear transfer'' part under Table 1), this already suggests that our method has better performance than a wide variety of commonly used losses. By results in [D] (Table 4), SimCLR achieves better performance than its supervised alternative while our approach outperforms SimCLR. We leave an extensive study on comparison to broader methods, such as [A, D] to future work.
>
> **Question 3**:  They also do not report performance on the training task of ImageNet.
>
> A3: We would like to bring to the reviewer's attention that this result has been reported in Table 3 before. In particular, in addition to the benefits on fine-grained classification, our method obtains better performance on ImageNet compared to cross-entropy loss (The approach CE is the fully-supervised baseline where the classifier weights are learned from scratch).
>
> **Question 4**: For long-tail, the method is not outperforming multiple methods of Table 2 on ImageNet-LT.
>
> A4: As discussed at the end of section 4, the TSC (Li et al., 2022) and KCL (Kang et al., 2020) use two-stage training involving supervised contrastive learning (SCL) to improve the performance of longtail classification. In addition, TSC and KCL are developed upon SCL, and SCL has the advantage of preserving the neighbor uniformity (discussed in the TSC paper) for features in the same class, which could benefit long-tail classification. However, the focus of our MSC features is on the diversity of features and has clearly benefited the fine-grained transfer learning under the context of neural collapse. Hence, the comparison to them is not fair, and we believe that our results can be further improved by introducing SCL in a similar way as TSC and KCL.

---

> > ### Author Response · Authors · 2022-11-16
> > **Response to Reviewer tRGD (2/2)**
> >
> >
> > **Question 5**: Fig 5c shows that setting the main hyper-parameter $\tau$ is far from trivial.
> >
> > A5: Like all other hyper-parameters in training a deep neural network, $\tau$ can simply be chosen via cross-validation. We note that $\tau$ is the only hyper-parameter introduced by our method and Fig 5c shows that the performance is not sensitive to the choice of $\tau$, hence our method is easy to use. For longtail learning, while our current strategy is to use the same $\tau$ for all minor classes, our method may further benefit from using different values of $\tau$ for different classes, depending on the amount of samples. While this may make the parameter tuning much more difficult, we note that this may be made easier and practical by choosing $\tau$ based on the class probability (similar idea was successfully used in prior work such as [Menon et al., 2020]). We leave an extensive study on this idea to future work.
> >
> > **Question 6**: it is unclear what model is used in each dataset and method in Table2.
> >
> > A6: We use ResNet32 for experiments on CIFAR10-LT and CIFAR100-LT, and ResNet50 for ImageNet-LT. More details are listed in the appendix Section B.3.
> >
> > **Question 7**: the "core" of the method contains unclear parts
> >
> > A7: We have provided definition of "cone" in Section 3.1 and would be happy to clarify if the reviewer can be more specific as to what part of the method is not clear.
> >
> > **Question 8**: It is unclear what Theorem 1 offers.
> >
> > A8: We explained this in the discussion following Theorem 1 and would be happy to clarify if the reviewer can be more specific as to what part of the explanation is not clear.
> >
> > **Question 9**: The analysis of NC comes from Papyan et al., 2020 in 2.2, and I don't see how this paper "examines the Neural Collapse phenomenon"
> >
> > A9: Papyan et al and followups study the emergence of NC, while our paper studies the question of whether NC is a good property or not. We have updated our writing in conclusion to clarify this point.
> >
> > **Question 10**: There are also implementation details that are missing in many places and therefore reproducibility is questionable.
> >
> > A10: Following the reviewer's comment, we have added the implementation details in appendix Section B to ensure the reproducibility.
> >
> > **Question 11**: Paragraph "Hinge Loss" in 3.2 is unclear in many places, while Eq9 is written in a very confusing way.
> >
> > A11: We thank the reviewer for bring this to our attention, we have updated Eq.9 as below in the paper.
> >
> > $$
> > h(\mathbf{u}_i, y_i, \boldsymbol{w}_k, b_k) =
> > \begin{cases}
> > h^+(\boldsymbol{u}_i, \boldsymbol{w}_k, b_k) \doteq \text{min} (<\boldsymbol{w}_k,\mathbf{u}_i\>, \text{stop-grad}(||\mathbf{u}_i||_2)||\boldsymbol{w}_k||_2 \cos \tau^+) + b_k \quad \text{if} \ k = y_i  \\\\
> > h^-(\boldsymbol{u}_i, \boldsymbol{w}_k, b_k) \doteq \text{max} (<\boldsymbol{w}_k,\mathbf{u}_i\>, \text{stop-grad}(||\mathbf{u}_i||_2)||\boldsymbol{w}_k||_2 \cos \tau^-) + b_k \quad \text{if} \ k \neq y_i
> > \end{cases}
> > $$
> >
> > Note that under the constraint that $||\boldsymbol{w}_k||_2=1,b_k=0~~\forall k \in$ {$1,...,K$}, the function can be further simplified. We would be happy to clarify if the reviewer have more questions and can be more specific as to what particular part of "Hinge Loss" is unclear.
> >
> > **Question 12**: Comments on first paragraph of intro and section 2.1.
> >
> > A12: Thanks for the suggestions. We have removed part of the description of loss function in section 2.1, and shortened the first paragraph in introduction.

---

> > > ### Comment · Reviewer_tRGD · 2022-11-19
> > > **Thank you for the response**
> > >
> > > I want to thank reviewers for their response.
> > >
> > > I still think that a) novelty is really hard to judge given that the ideas that compose the proposed approach are very close to others in the related works [A-D] and there is no real direct comparison to those; b) there are missing experiments for both fine-grained and long-tail (both areas are in the paper title) and the method does not outperform other simple methods for LT; c) the paper still text lacks clarity after very minor clarifications.
> > >
> > > Some more detailed notes on their responses:
> > > Q1/Q2: I get that the proposed approach is simpler, yet, you should be experimentally comparing to the closest related works, not just discussing them; especially [C] also has the same complexity. I am sorry but this is missing  to support your claims.
> > >
> > > Q3: Thank you for pointing out Table 3 to me, I had missed it upon first read.
> > >
> > > Q4: the methods you mention, especially Kang et al 2020 is very simple modifications that give really good results on LT. If your claim, which is in the title, is that the proposed approach is suited for LT recognition, you have to not compare to simple baselines but some more advanced baselines tailored to the task.
> > >
> > > Q6-9: Thank you to the authors for making some small edits to the text; however, the paper text still lacks a lot in clarity to me especially in the core section.

---

> > > > ### Author Response · Authors · 2022-12-01
> > > > **New Response (1/2)**
> > > >
> > > > We thank the reviewer for the response.
> > > >
> > > > > novelty is really hard to judge given that the ideas that compose the proposed approach are very close to others in the related works [A-D].
> > > >
> > > > While our method is indeed related to those in [A-D], we do not agree with the assessment that our method is “very close to” them.
> > > >
> > > > - Our method is related to [A, B] in the sense that both [A, B] and us present methods for learning diverse representations. However, we have a drastically different approach for obtaining diversity. The approaches in [A, B] are based on k nearest neighbor search and the training pipelines are complex. In contrast, our method is to directly hand-craft the geometry of the features (i.e., the maximal-separating-cones) for obtaining diversity. Such a novelty is of great significance because it offers
> > > >
> > > >   1. explainability, as it is very clear what features we are learning and using for downstream tasks, as well as how the hyper-parameter controls the property of interest (i.e., diversity), and
> > > >
> > > >   2. simplicity, as our method requires a very small change to the commonly used pipeline.
> > > >
> > > >   Importantly, this is achieved because our method is deeply grounded in a recent line of work on Neural Collapse, that provides solid empirical and theoretical results for demystifying deep features. Our method is first-of-its-kind to develop new methods based on neural collapse, and may inspire the development of other geometry for obtaining other desired properties as well.
> > > >
> > > > - Regarding a comparison to [C], the *only* similarity is that both [C] and us use a fix the classifier. All the other aspects are completely different:
> > > >
> > > >   1. the classifier that [C] uses is the orthogonal vectors, while we use simplex ETFs;
> > > >   2. the goal of [C] is to show that the classifier can be fixed, while we aim to design a feature geometry;
> > > >   3. [C] does not redesign the loss to encourage diversity;
> > > >   4. [C] did not test their method for longtail and fine-grained classification.
> > > >
> > > > - Perhaps the *only* similarity between our paper and [D] is that both of them present new loss functions for supervised learning. On the other hand, all aspects of the loss functions in these two papers are completely different, such as their motivation, design method, training procedure, and properties of the learned representation.

---

> > > > > ### Author Response · Authors · 2022-12-01
> > > > > **New Response (2/2)**
> > > > >
> > > > >
> > > > > > ... there is no real direct comparison to those ([A-D])
> > > > >
> > > > > Per the reviewer's request, we offer a direct comparison of our method to [A - D] as follows.
> > > > >
> > > > > - We conduct experiments on CIFAR-20 & CIFAR-100 to compare our HingeCE with [C]. We set ResNet-32 as feature backbone and implement [C] to reproduce the results. However, as the approach is sensitive to temperature, the test accuracy on CIFAR-100 is slightly lower, i.e., 75.5\% by [C] and 76.8\% by CE baseline.
> > > > > Then, we perform pre-training on CIFAR-20 and then train a linear classifier for CIFAR-100. As compared below, since [C] only fixes the classifier but does not consider the variability collapse issue, it achieves similar performance with CE baseline on CIFAR-20 but the linear evaluation accuracy on CIFAR-100 is 34.2\%. However, as our approach preserves the diversity within each class, Hinge CE improves the linear evaluation accuracy to 45.7\% without hurting the test accuracy on CIFAR-20.
> > > > >
> > > > >   | Approach | CIFAR-20 Test Acc. | CIFAR-100 Linear Eval. |
> > > > >   |----------|------------------------|------------------------|
> > > > >   | CE       | 82.7                  | 33.5                  |
> > > > >   | [C]      | 82.2                  | 34.2                  |
> > > > >   | HingeCE  | 82.8                  | 45.7                  |
> > > > >
> > > > > - Then, we compare our HingeCE with LOOK [A] where no strong augmentation is applied during ImageNet pre-training. The results of [A] are reported in [A] (Table 1) where the four datasets are all about fine-grained classification and are evaluated by both LOOK [A] and HingeCE. On most of the datasets, HingeCE can consistently outperform LOOK. Regarding the accuracy difference on Aircraft and Cars, we think the main reason is that the biases learned in the two approaches are not exactly the same. However, our approach still achieves higher accuracy on average.
> > > > >
> > > > >   | method | Aircraft | Car | Flower | Pet | Mean |
> > > > >   |:------:|:--------:|:---:|:------:|:---:|:----:|
> > > > >   | LOOK [A] |   56.8   | 69.2 |   94.9  | 90.1 | 77.8 |
> > > > >   |  HingeCE |   61.2   | 62.2 |   97.8  | 92.2 | 78.4 |
> > > > >
> > > > > - For a comparison with [D], it can be seen from [D] (Table 4) that its method underperforms SimCLR on linear evaluation; meanwhile, our method outperforms SimCLR as shown in Table 1. Hence, our method outperforms the method in [D].
> > > > >
> > > > > - Finally, we note that it is very difficult to compare with [B] since 1) [B] does not report the transfer accuracy of linear evaluation, so we cannot directly compare with their reported results, 2) the implementation of [B] is not trivial, e.g.,
> > > > >
> > > > >   1. huge memory storage where the features of all 1.2M images in ImageNet train set should be stored for experiments,
> > > > >   2. exponential moving average is used for network update,
> > > > >   3. two losses (instance loss and kNN loss) are combined while hyper-parameters such as temperature and weights are not given,
> > > > >
> > > > >   and 3) we do not find code release for [B]. More importantly, we would like to re-emphasize that [B] does not study the features used for downstream space directly, i.e., [B] employs a projector and supervises the feature learning of projector output while the features for downstream tasks (i.e., input features of the projector) are not clearly studied.
> > > > >
> > > > > > the method does not outperform other simple methods for LT ... the methods you mention, especially Kang et al 2020 is very simple modifications that give really good results on LT
> > > > >
> > > > > We would like to bring to the reviewers attention that our method does outperform the KCL on two datasets, CIFAR10-LT and CIFAR-100-LT. In particular, the performance improvement is by a large margin (around 3%).
> > > > >
> > > > > Our method only underperforms KCL on a single dataset, ImageNet-LT, by a relatively much smaller margin (around 0.7%). However, please note that this comparison is not fair at all. KCL employs both color jittering and flipping as data augmentation for training, but we only use image flipping. To understand the effect of jittering, we used the official KCL github repo and obtained its performance as 51.2\%. Then, we remove the jittering for KCL during training and its performance drops to 49.6\%, which is lower than our HingeCE (50.8%). Hence, under a fair comparison, our method is also better than KCL on ImageNet-LT.
> > > > >
> > > > >
> > > > > Moreover, the main message of our paper, namely within-class diversity improves longtail learning performance, is orthogonal to the main message of Kang et al 2021, namely balanced class separation improves longtail learning performance. It is possible that combining the methods from these two papers can further improve the performance. This could be a good topic for a follow-up work.
> > > > >
> > > > > > the paper text still lacks a lot in clarity to me especially in the core section.
> > > > >
> > > > > We kindly ask the reviewer to provide more details on which particular part of the paper is unclear. For example, what is the ``core'' section that the reviewer is referring to? We would be very happy to improve the clarity of the writing based on the reviewer's feedback.

---

### Decision · Program_Chairs · 2023-01-20

**Decision:**

Reject

**Justification For Why Not Higher Score:**

Multiple reviewers express concerns about novelty, the motivation for the proposed approach, the clarity of presentation, and the quality of the experimental validation of the work.

**Justification For Why Not Lower Score:**

N/A

**Metareview: Summary, Strengths And Weaknesses:**

The paper studies a method that tries to prevent model representations of instances with the same class from collapsing when the model is trained using a classification objective. The paper argues that such collapse is undesired when class labels are coarse-grained but we also wish to transfer them to finer-grained classification tasks.

Initially, two reviewers were lukewarm about the paper. The reviewers express concerns about novelty, the motivation for the proposed approach, the clarity of presentation, and the quality of the experimental validation of the work. Some of these concerns were addressed in the discussion, but these reviewers find that there are still missing comparisons. Also, they find the clarity of presentation lacking even in the revised manuscript.

One reviewer (6L41) provided a more positive overall assessment of the paper, but did not provide much context on what they liked about the paper. The AC connected with reviewer 6L41 offline to get more context on their assessment. Reviewer 6L41 let the AC know that, on second thought, they may have overrated the paper a bit and that they share some of the concerns by the other reviewers around novelty and motivation.

**Summary Of Ac-Reviewer Meeting:**

Due to lack of time, I connected with the positive reviewer (6L41) offline. They let me know that they feel their initial assessment was off.